

# Establishing an optimized method for the separation of low and high abundance blood plasma proteins

Henian Yang[1], Guijie Wang[1], Tiantian Zhang[2], John H. Beattie[3] and Shaobo Zhou[1]

[1] School of Life Sciences, Institute of Biomedical and Environmental Sciences and Technology (iBEST), University of Bedfordshire, Luton, Bedfordshire, UK
[2] School of Applied Sciences, Bournemouth University, Bournemouth, UK
[3] The Rowett Institute, University of Aberdeen, Aberdeen, UK

## ABSTRACT

The study tested the efficiency and reproducibility of a method for optimal separation of low and high abundant proteins in blood plasma. Firstly, three methods for the separation and concentration of eluted (E: low abundance), or bound (B: high abundance) proteins were investigated: TCA protein precipitation, the ReadyPrep™ 2-D cleanup Kit and Vivaspin Turbo 4, 5 kDa ultrafiltration units. Secondly, the efficiency and reproducibility of a Seppro column or a ProteoExtract Albumin/IgG column were assessed by quantification of E and B proteins. Thirdly, the efficiency of two elution buffers, containing either 25% or 10% glycerol for elution of the bound protein, was assessed by measuring the remaining eluted volume and the final protein concentration. Compared to the samples treated with TCA protein precipitation and the ReadyPrep™ 2-D cleanup Kit, the E and B proteins concentrated by the Vivaspin4, 5 kDa ultrafiltration unit were separated well in both 1-D and 2-D gels. The depletion efficiency of abundant protein in the Seppro column was reduced after 15 cycles of sample processing and regeneration and the average ratio of $E/(B + E) \times 100\%$ was $37 \pm 11(\%)$ with a poor sample reproducibility as shown by a high coefficient of variation (CV = 30%). However, when the ProteoExtract Albumin/IgG column was used, the ratio of $E/(B + E) \times 100\%$ was $43 \pm 3.1\%$ ($n = 6$) and its CV was 7.1%, showing good reproducibility. Furthermore, the elution buffer containing 10% (w/v) glycerol increased the rate of B protein elution from the ProteoExtract Albumin/IgG column, and an appropriate protein concentration (3.5 µg/µl) for a 2-D gel assay could also be obtained when it was concentrated with Vivaspin Turbo 4, 5 kDa ultrafiltration unit. In conclusion, the ProteoExtract Albumin/IgG column shows good reproducibility of preparation of low and high abundance blood plasma proteins when using the elution buffer containing 10% (w/v) glycerol. The optimized method of preparation of low/high abundance plasma proteins was when plasma was eluted through a ProteoExtract Albumin/IgG removal column, the column was further washed with elution buffer containing 10% glycerol. The first and second elution containing the low and high abundance plasma proteins, respectively, were further concentrated using Vivaspin® Turbo 4, 5 kDa ultrafiltration units for 1 or 2-D gel electrophoresis.

Corresponding authors
John H. Beattie, J.Beattie@abdn.ac.uk
Shaobo Zhou,
shaobo.zhou@beds.ac.uk

## INTRODUCTION

Blood plasma is an easily available bio-fluid and is therefore routinely used for monitoring changes in protein levels which may be actively secreted or leak from cells throughout the body. The highest abundance proteins in blood plasma are albumin, globulins and fibrinogen which comprise about 60%, 30% and 4% of whole plasma proteins (7–8 g/dL), respectively (*Farrugia, 2010*; *Anderson & Anderson, 2002*). Among the remaining proteins, about 1%, are regulatory and they are comprised of thousands of low abundance proteins, such as enzymes, proenzymes and hormones (*Anderson & Anderson, 2002*). Plasma protein biomarkers of disease progression is currently a very active research area (*Zhang et al., 2013*).

As a common metabolic pool, plasma has been a very important material for biomarker discovery (*O'Connell, Horita & Kasravi, 2005*; *Zhang et al., 2013*). Biomarker targets of disease progression are typically found at low concentrations (*Anderson & Anderson, 2002*) and so identification and quantification of these proteins is challenging. Two-dimensional gel electrophoresis (2-DE) with IPGs were developed in the 1970s (*Görg et al., 2009*) but it still has many challenges (*Zhou et al., 2005*), while LC–MS based proteomics has been a more recent development (*Kitteringham et al., 2009*; *Van den Broek, Niessen & Van Dongen, 2013*). They have both proved useful in plasma biomarker discovery. However, the plasma proteome has a large dynamic range of individual protein concentrations (10 orders of magnitude). Therefore, there are several barriers to overcome for identification and quantification of low abundance proteins of interest using 2-DE and LC–MS. One of them is the visualization and measurement of lower-abundance proteins which are typically masked by the highly-abundant proteins in a standard measurement (*Kovacs & Guttman, 2013*; *Boschetti & Righetti, 2009*; *Pernemalm et al., 2009*; *Anderson & Anderson, 2002*). In order to tackle this problem, several commercial columns have been developed to deplete the higher abundance proteins. These columns have helped to further the research process, even though there are several significant challenges to overcome, for example, sample reproducibility.

Trichloroacetic acid (TCA) can be added to too diluted protein samples in order to precipitate and concentrate proteins or remove salts and detergents to clean the samples. TCA precipitation was frequently used to prepare samples for SDS-PAGE or 2D-gels (*Zhou et al., 2005*; *Koontz, 2014*). The ReadyPrep 2-D cleanup kit was developed by Bio-rad. It has similar mechanisms to TCA precipitation, using a modified traditional TCA-like protein precipitation to remove ionic contaminants, for example, detergents, lipids and phenolic compounds, from protein samples to improve the 2-D resolution and reproducibility (*Posch, Paulus & Brubacher, 2005*). Vivaspin® Turbo 4 can handle up to four ml sample and ensures maximum process ultra-fast speed down to the last few micro liters after > 100 fold concentration with high retentive recovery > 95%. In addition,

it has universal rotor compatibility and easy recovery due to a unique, angular and pipette-friendly dead-stop pocket (*Capriotti et al., 2012*).

The Seppro Column (SEP130-1KT; Sigma–Aldrich Ltd., St. Louis, MO, USA), specified for rat plasma, is designed to remove seven highly abundant proteins: albumin, IgG, fibrinogen, transferrin, IgM, haptoglobin and alpha1-antitrypsin. The column contains an antibody-coated resin, and this depletion technology uses a mixture of small single-chained recombinant antibody ligands along with conventional affinity purified polyclonal antibodies. The efficiency of high-abundance protein depletion is 90%. Following depletion of these high abundance proteins, the remaining lower abundance proteins were then loaded at a 20–50 times higher concentration in a 2-DE or LC separation. Seppro Columns have also been used for human plasma (*Corrigan et al., 2011*; *Polaskova et al., 2010*) and plant proteins (*Cellar et al., 2008*). Most studies have focused on their binding efficiency, which is reported to be high for the targeted abundant proteins; however, there is no report on the reproducibility of abundant proteins depletion with repeated use of the columns, even though it has been claimed by the manufacturer that columns can be used up to 100 times. High efficiency and good reproducibility are important in maintaining a reproducible protein profile. Some animal or human nutritional interventions, such as zinc depletion (*Kwun et al., 2009*; *Ou et al., 2013*; *Watanabe et al., 2010*), can affect hundreds of plasma proteins which may be found in high or low concentrations. Thus, related biomarker discovery has to focus on both abundant and low abundance proteins separately.

Another frequently used column, ProteoExtract^TM Abundant Protein Removal Kit, is a disposable column, which was developed in 2004 for the purpose of enhancing low abundance protein resolution. It only removes two abundant proteins, albumin and IgG. However, it is highly specific, exhibits little to zero nonspecific binding and uses a combination of an albumin-specific resin and a unique immobilized protein A polymeric resin. It has been tested for different purposes (*Olver et al., 2010*; *Sawhney, Stubbs & Hood, 2009*; *Liang et al., 2007*; *Björhall, Miliotis & Davidsson, 2005*). Even though several studies have cited its use, there is no report on reproducibility between individual columns, especially for the recovery of bound protein from the column.

This is the first systematic study to investigate: (1) several methods for sample preparation, for example, appropriate conditions for depletion, desalting and concentration of protein samples; (2) the reproducibility and depletion efficiency of abundant protein removal from plasma samples using a Seppro column repeatedly or using the individual ProteoExtract^TM Abundant Protein Removal Kit; and (3) the development of an efficient elution buffer to wash out the bound protein from the ProteoExtract Albumin/IgG column.

## METHOD AND MATERIALS

### Chemicals

Seppro rat spin column (Seppro^® Rat. SIGMA/SEP130; Sigma–Aldrich Ltd., St. Louis, MO, USA), Laemmli buffer, SDS, glycerol, Tris base, bicinchoninic acid (BCA) protein assay (All from Sigma–Aldrich Ltd., Dorset, UK); ProteoExtract Albumin/IgG removal kit

(Cat: 122642; Merck, Kenilworth, NJ, USA); the BCA protein assay kit (The Thermo Scientific Pierce, Waltham, MA, USA); rat plasma.

## Animals

Male Hooded Lister rats ($n$ = 50, body weight was 200 ± 25 (g)) were given semi-synthetic egg white-based diets, containing zinc from <1, up to 35 mg Zn/kg. The rats were handled and studied in compliance with the UK Animals (Scientific Procedures) Act 1986 with appropriate licensing. Rats were individually housed in polypropylene cages with a 12h:12h light:dark cycle and a room temperature of 22–24 °C. Blood was collected and plasma was isolated using our established protocol (*Posch, Paulus & Brubacher, 2005*). The study was approved by the Small Animal Ethics Committee of University of Aberdeen (The Rowett Institute, University of Aberdeen, the reference was 604012) and monitored by qualified university-based veterinary surgeons.

## Depletion of abundant plasma proteins using a Seppro column

The methods, mainly based on the protocols provided by the company (Sigma–Aldrich, Dorset, UK), as well as a modified procedure, is described in the Supplemental Information 1.

## Depletion of abundant plasma proteins using ProteoExtract Albumin/IgG column

The detailed procedure of column equilibration and sample treatment using ProteoExtract Albumin/IgG removal kit is shown in Supplemental Information 2. The procedure was briefly as follows: (1) Collection of the low abundant proteins: The column was inverted on tissue paper for 5 min; 850 µL of the binding buffer was added into the column and allowed to pass through the resin bed by gravity flow; a 40 µL sample of rat plasma was then diluted with 360 µL of binding buffer and applied onto the column. The diluted sample was then allowed to pass through the resin bed by gravity-flow; 600 µL of binding buffer was added to wash the column by gravity-flow and the eluent, which contained the low abundance proteins, was collected; (2) Collection of the high abundance proteins: one ml of Laemmli buffer (50 ml Laemmli buffer containing 0.3785 g Tris base, (62.4 mm); 1.0279 g SDS (2%); buffer 1 containing glycerol 9.92 ml (25% w/v) or buffer 2 containing glycerol 3.97 ml (10% w/v), pH 6.8) in a tube was left in a boiling water bath for 5 min, then cooled to room temperature. A total of 850 µL of above Laemmli buffer was added onto the column and allowed to pass through, by gravity-flow. This step was repeated again for further washing to elute the bound protein. The eluent contained the abundant proteins.

## Concentration, desalting, clean up and quantification of eluted or bound proteins

Three methods were compared in order to concentrate the eluted plasma fraction and the bound proteins to at least five µg/µl for the generation of quality 2-D gels. They were the Vivaspin Turbo 4, 5 kDa ultrafiltration unit (VS04T11; Sartorius, Epsom, UK), TCA

precipitation (*Zhou et al., 2005*) or the ReadyPrepTM 2-D cleanup Kit (Catalog #163-2130; Bio-Rad, Hercules, CA, USA). A detailed protocol for each is provided in Supplemental Information 3. Samples were desalted using pH 7.4 50 mm Tris buffer. Protein quantification was achieved using the BCA method, or the RC-DC protein assay (Thermo Scientific Pierce, Waltham, MA, USA) (Supplemental Information 4). The impact of each concentration method on the quality of the protein separation profile was analyzed by loading the samples onto either 1-D or 2-D gels.

## Reproducibility of the Seppro spin column and the ProteoExtract Albumin/IgG removal kit for the depletion of abundant proteins

Twenty seven plasma samples were depleted of abundant proteins sequentially using a Seppro spin column. Every other six samples, one quality control sample was used to perform the same depletion procedure. The eluted (E: low abundance) and bound (B: abundant) protein was quantified. The ratio of either E or B protein to total (E + B) recovered proteins during 27 sample as well as the four quality control sample depletions was used to assess the stability and reproducibility of the Seppro spin column. Plasma (40 μL, $n = 6$) samples were diluted with 360 μL of binding buffer and were loaded onto a ProteoExtract Albumin/IgG removal column separately. The amount of E and B protein were analyzed. The ratio of their amount to the total protein previously depleted was calculated and their CV% was used to assess the reproducibility of the Albumin/IgG removal kit.

## One and two-dimensional SDS PAGE electrophoresis
### One-dimensional SDS PAGE
Protein (15 μg) was loaded in each well of a 4–12% Bis–Tris Criterion™ XT precast gel (Catalog 345-0124; Bio-Rad, Hercules, CA, USA). The electrophoresis was performed using a Criterion™ cell (Catalog 165-6001; Bio-Rad, Hercules, CA, USA) with MOPS running buffer (Catalog 161-0788; Bio-Rad, Hercules, CA, USA) with a 200 V constant power supply. When the bromophenol blue ran to the bottom of the gel, the power supply was turned off and the gel was removed and stained with Coomassie Blue.

### Two-dimensional gel electrophoresis
Protein samples (200 μg) were diluted with the buffer (7 M urea, 2 M thiourea, 4% CHAPS, 2.0% bio-lyte, 3/10 ampolyte) to a volume of 325 μl, and 15 μl of 3.5% DTT was then loaded onto a 18 cm IPG readystrips (Catalog 163-2007; Bio-Rad, Hercules, CA, USA) with a linear pH gradient of 3–10 by passive in-gel rehydration. Rehydration was performed at 20 °C for 1 h without applied voltage on an IEF cell (BioRad, Hercules, CA, USA). Then mineral oil was added onto the strip. The rehydration took an extra 16 h (50 V/strip). The strips were transferred to a clean tray, with a paper wick wetted with 10 μl of ddH$_2$O placed at the anode end and a wick wetted with 15 μl of 3.5% DTT placed at the cathode end. The strip was then overlaid with mineral oil and the initial startup and ramping protocol followed as per the instruction booklet for the IEF cell. After 1 h, the strip was removed to a tray containing fresh wicks and overlaid with mineral oil. The run

proceeded until the preset volt-hours value had been reached, after which the voltage was maintained at 500 V until the strip was ready to be transferred to the second dimension SDS-PAGE. IPG strips were removed from the focusing tray and reduced by equilibrating the strips side up, in a solution (three ml) (containing 6 M urea, 2% (w/v) SDS, 20% (v/v) glycerol, 375 mm Tris-HCl (pH 8.8), 130 mm DTT) for 13 min at room temperature with gentle agitation, before being alkylated in a solution (three ml) (containing 6 M urea, 2% (w/v) SDS, 20% (v/v) glycerol, 375 mm Tris-HCl (pH 8.8), 135 mm Iodoacetamide) for 13 min at room temperature. The strip was trimmed from both the anodic and cathodic ends to 15.5 cm and applied to the top of a 18 ×18 cm gel cassette (8–16% cast gels) with the lower pH end of the strip to the extreme left and then overlayed with molten agarose (2%, w/v) in DALT tank buffer (24 mm Tris base, 200.5 mm glycine and 0.1% (w/v) SDS containing 2 mg/100 ml bromophenol blue). The second dimension separation was performed in a Hoefer ISO DAKT tank (Bio-Rad, Hercules, CA, USA) filled with the DALT tank buffer. The gels were typically run at 200 V for 9.5 h or until the bromophenol blue front reached the bottom of the gel. A holding voltage of 50 V was applied after the gel run to prevent diffusion.

## Coomassie blue stain

The gels were fixed in 200 ml solution (50% (v/v) ethanol, 2% (v/v) ortho-phosphoric acid, 48% $H_2O$) for 3 h and washed with $H_2O$ for at least 1 h with a couple of changes for rehydration. They were then stained with 200 ml Coomassie blue (34% methanol, 2% ortho-phosphoric acid, 64% $H_2O$, containing 1 mg/1 ml Coomassie blue) for three days, according to the manufacturer's instructions. The gels were then scanned with the GS-800 Calibrated Densitometer and analyzed using Progenesis SameSpots (Nonlinear, UK).

## Statistics

After normalization and spot matching on Progenesis SameSpots, the normalized volume (densities) of all matched spots were statistically analyzed using Genstat (VSN International, Hemel Hempstead, UK).

## RESULTS

### Protein recovery rate of three concentration methods and their protein profiles in 1-D and 2-D gels

The protein recovery rate of three methods using the TCA precipitation, the ReadyPrep[TM] 2-D cleanup Kit and the Vivaspin Turbo 4, 5 kDa ultrafiltration unit, to concentrate the samples, was 67, 68 and 56% respectively. Samples treated with the ReadyPrep[TM] 2-D cleanup Kit were too dilute (2.62 μg/μl, $n = 3$) for loading onto 2-D gels, compared to the higher protein concentration (3.27 μg/μl, $n = 3$) obtained after Vivaspin Turbo 4, 5 kDa ultrafiltration unit sample processing. When the crude plasma samples were treated with the Seppro spin column, total protein recovery rate was 55.3% which included 27.7% of eluted protein, 2.4% nonspecifically bound protein and 25.2% bound protein. The recovery rate after ProteoExtract Albumin/IgG removal kit treatment was 58% ($n = 6$), containing 33.6% eluted protein and 24.4% abundant protein. Both columns generated

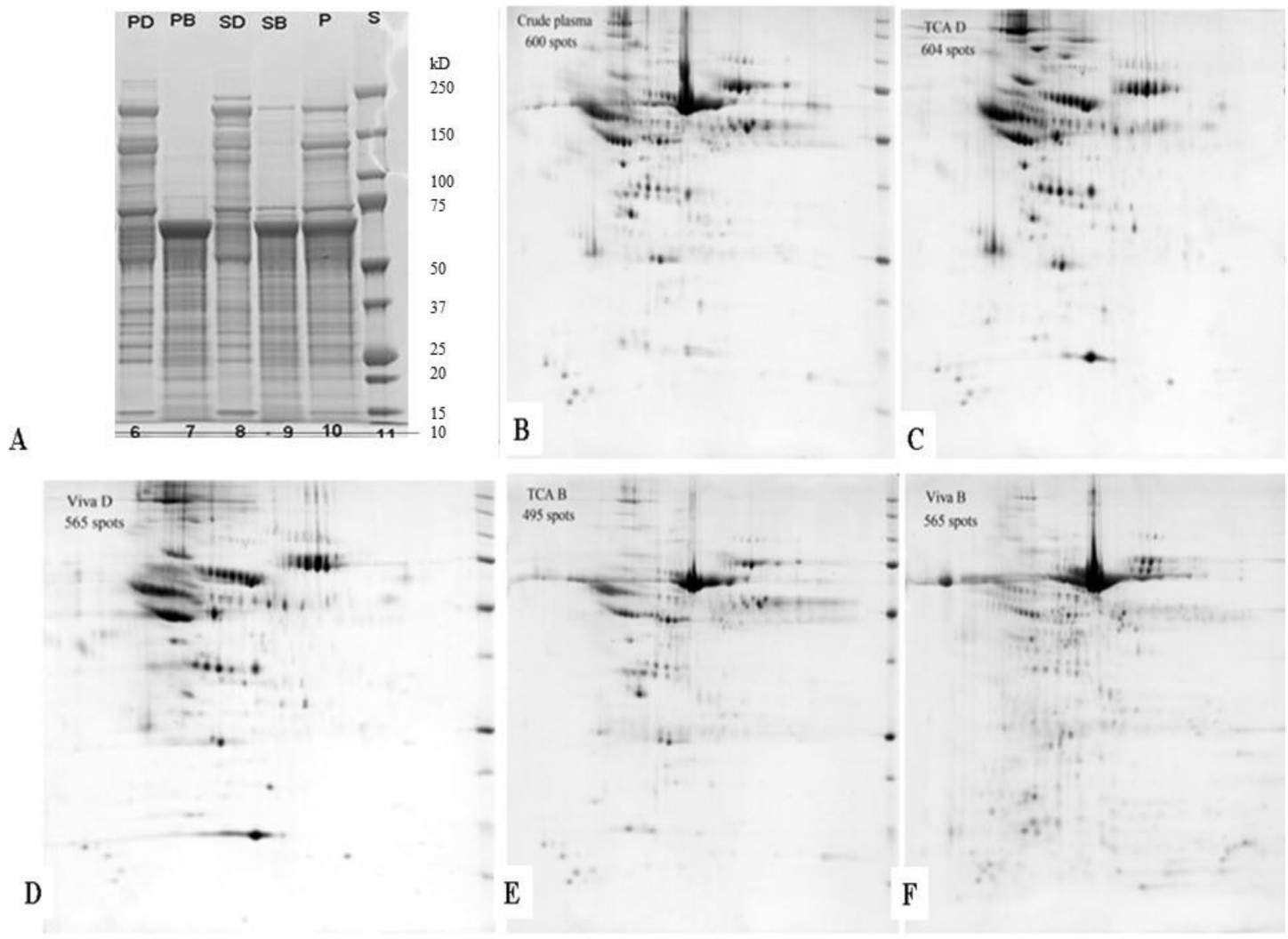

**Figure 1 Gel images of 1 or 2-D gels of proteins from different treatments.** (A) 1-D gel image of 15 μg of eluted and bound proteins prepared by ProteoExtract Albumin/IgG removal kit and Seppro column kit. The sample concentration was adjusted to 3.5 μg/μl with 50 mm tris-HCl pH 7.4 before treatment with sample buffer. S: precision plus protein dual color standards. PD, ProteoExtract Albumin/IgG removal column eluted protein; PB, ProteoExtract Albumin/IgG removal column bound protein; P, crude plasma; SD, Seppro column eluted protein; SB, Seppro column bound proteins. (B–F) 2-D images of protein samples precipitated with TCA or concentrated by Vivaspin Turbo 4, 5 kDa ultrafiltration unit. A total of 200 μg protein was loaded onto an 18 cm pH 3–10 IPG strip and an 8–16% gradient SDS PAGE gel in the second dimension. The gels were stained with Coomassie blue; (B) crude rat plasma protein; (C) eluted protein was precipitated with TCA; (D) eluted protein was concentrated by Vivaspin Turbo 4, 5 kDa ultrafiltration unit; (E) bound protein was precipitated with TCA and (F) bound protein was concentrated by Vivaspin Turbo 4, 5 kDa ultrafiltration unit.

similar results for the protein recovery rate. Samples obtained from separation of plasma on ProteoExtract Albumin/IgG removal kits and Seppro spin column were run on a 1-D gel (Fig. 1A), to compare the difference between two column separations. The influence of sample concentration by TCA precipitation and Vivaspin Turbo 4, 5 kDa ultrafiltration unit on protein separation was studied by loading two samples from each separation into two 18 cm 2-D SDS-PAGE gels (Figs. 1B–1F). Samples obtained from ProteoExtract Albumin/IgG removal kit were run onto 2-DE gels (Fig. 2).

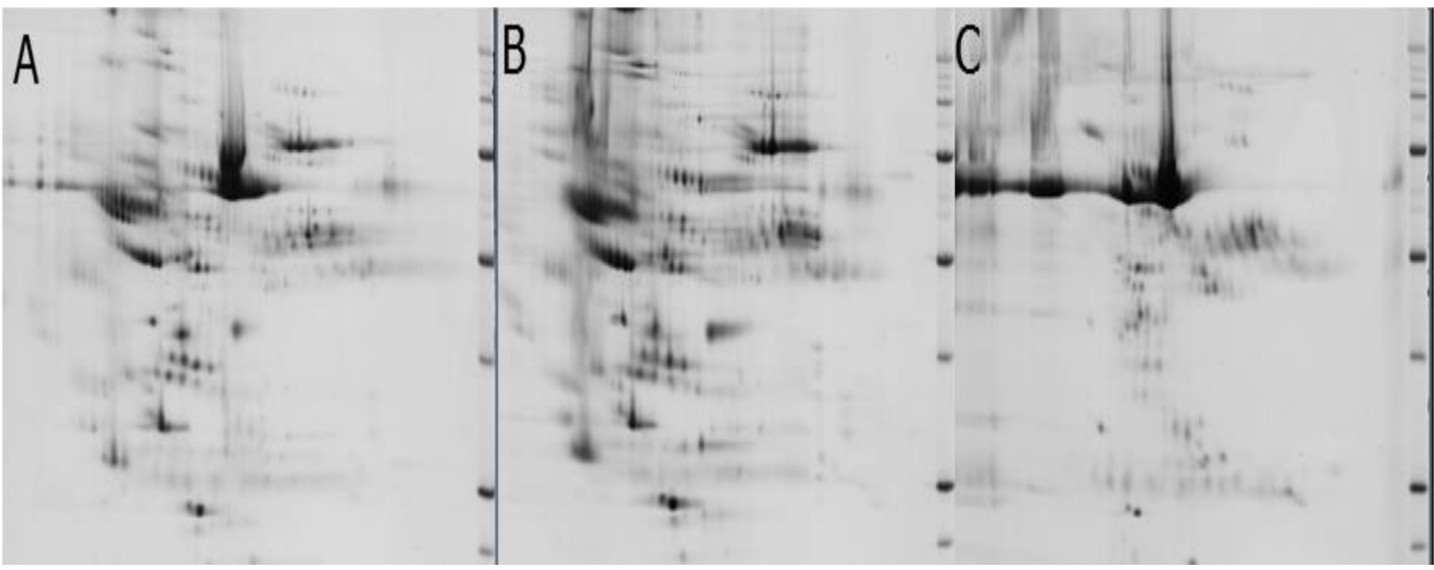

**Figure 2 2-D gel images of plasma protein samples eluted from a ProteoExtract Albumin/IgG removal kit and concentrated using Vivaspin Turbo 4, 5 kDa ultrafiltration unit.** A total of 200 μg protein was loaded onto an 18 cm pH 3–10 IPG strip and an 8–16% gradient SDS PAGE gel was used in the second dimension. The gels were stained with Coomassie blue. (A) crude rat plasma protein; (B) eluted protein; (C) bound protein.

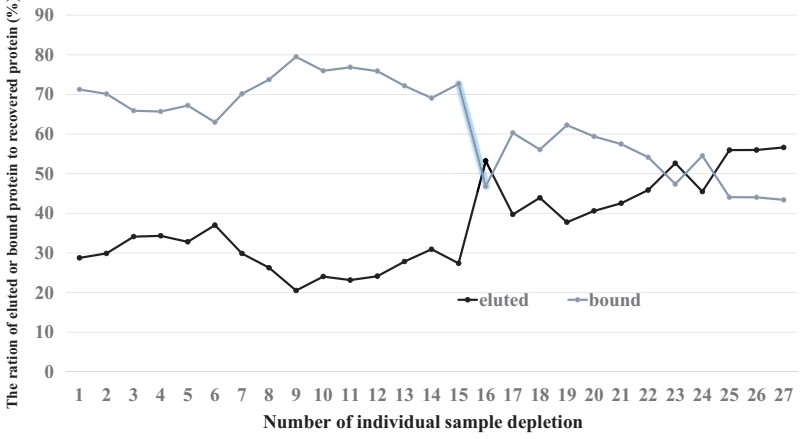

**Figure 3 The proportion of eluted (E) and bound (B) protein obtained with repeated depletion-regeneration Seppro spin column cycles for plasma sample depletion.**

## Reproducibility of the Seppro spin column and the ProteoExtract Albumin/IgG removal kit in depleting abundant proteins

Results for protein separation in 1-D and 2-D of both "Protein recovery rate of three concentration methods and their protein profiles in 1-D and 2-D gels" showed efficient depletion of abundant proteins when using the Seppro column. The ratio of eluted protein (E) or bound protein (B) to total recovered proteins (E + B) during repeated depletion of 27 samples is shown in Fig. 3. The trend in ratio is an indication of durability and reproducibility of the column for repeated depletion and regeneration. The average ratio of

**Table 1 Reproducibility of the separation of eluted (low abundance) and bound (abundant) plasma proteins using the ProteoExtract Albumin/IgG removal kit before and after concentration.**

| | Eluted (low abundance) proteins | | | Bound (abundant) protein | | |
|---|---|---|---|---|---|---|
| | Amount (μg) | CV (%) | Eluted proteins to total protein loaded (%) | Amount (μg) | CV (%) | Bound protein to total protein loaded (%) |
| Before concentration* | 967 ± 68.2 | 7.1 | 43 ± 3.1 | 668 ± 66.2 | 9.9 | 28 ± 3.0 |
| After concentration* | 749 ± 40.5 | 5.4 | 33 ± 1.8 | 534 ± 49.5 | 9.3 | 24 ± 2.2 |

Notes:
* Calculated by the total volume × concentration.
CV, coefficient variation. Amount of protein loaded onto the column was 2,215 μg ($n = 6$).

E/(E + B) ×100% was 37 ± 11 (%, $x \pm$ SD, $n = 27$) with a bigger CV which was 30%. There was a variety change during the whole procedure, for example, initially relatively stable for the first fifteen depletions at around 28% but then increased dramatically up to 53% at 16th depletion which may be caused by transferring the resin to a new column which is required after several depletions. Then it reduced to average of 41% in the following six extra depletions, then further increased to 53% in the following five depletions. The average ratio of the eluted protein compared to total protein for 27 samples was 63% ± 11 (%, $x \pm$ SD, $n = 27$) and with a CV of 18%. Using the ProteoExtract Albumin/IgG removal column, the average recovery rate of eluted proteins (amount of protein in the eluted solution /total protein loaded onto the column) was 43% ($n = 6$), while the recovery rate of abundant proteins (amount of bound protein on the column /total protein loaded onto the column) was 33% ($n = 6$). Their CVs for the recovery rate of eluted and abundant proteins were 7.1% and 5.4% respectively (Table 1).

### The efficiency of two eluted buffers for remove bound proteins from the column

Laemmli buffer containing with 25% or 10% glycerol was used to wash out bound proteins from column. The solutions collected were further concentrated using Vivaspin Turbo 4, 5 kDa ultrafiltration unit respectively. The total volume of the solution remained during the concentration procedure is shown in Fig. 4.

## DISCUSSION

### Concentration methods and protein recovery

The quality of 2-D gels using of samples prepared from both TCA and Vivaspin Turbo 4 were similar in their sharpness resolution and clearance clarity (Figs. 1C and 1E for eluted proteins, 1D and 1F for bound proteins). However, samples treated with TCA precipitation did not generate higher spots numbers for bound protein samples compared to the samples prepared by Vivaspin Turbo 4 (495 and 659), even though the number of eluted protein spots from TCA- precipitation treated samples was 39 spots higher than that from the Vivaspin Turbo 4 prepared eluted protein sample (604 and 565) (Protein recovery rate of three concentration methods and their protein profiles in 1-D and 2-D gels). This might be caused by the TCA -prepared bound sample which had a lower

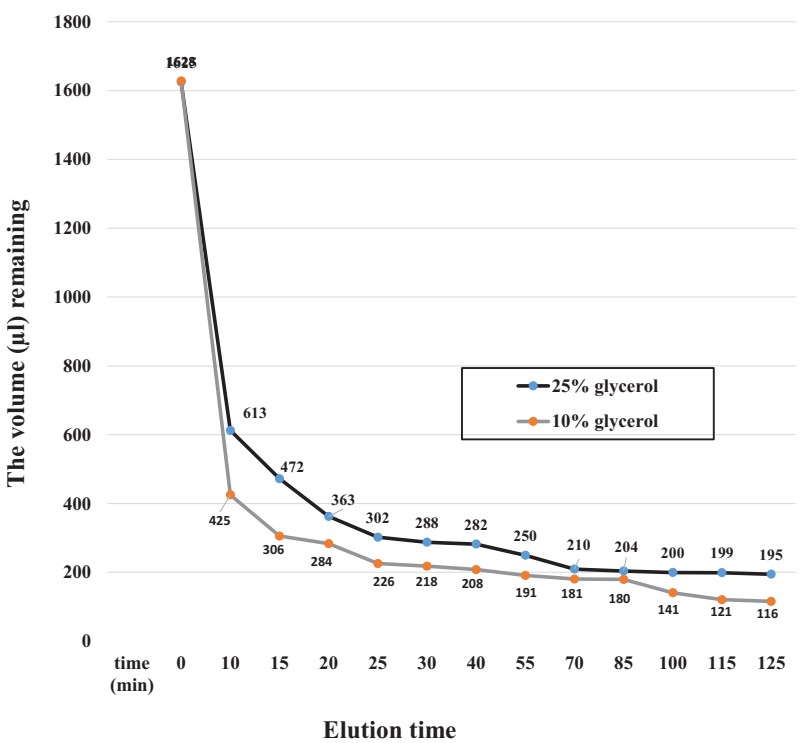

**Figure 4 The change in elution volume with time during Vivaspin Turbo 4, 5 kDa ultrafiltration unit concentration of fractions containing 10% and 25% glycerol.**

separation efficiency. Based on above results, the Vivaspin Turbo 4, 5 kDa ultrafiltration unit was selected to concentrate protein samples in the study. From Fig. 1A, clear and dense bands about 66 kDa were observed in bound proteins, collected from both columns. However, several bands above 75 kDa were observed in the samples of using Seppro spin column. Clearly, eluted proteins prepared by the Seppro spin column had more bands generated in the gel. Also, some dense bands around 140 kDa and 190 kDa had been lightly stained, compared to the sample prepared by the ProteoExtract Albumin/IgG removal kit. Comparison of the untreated plasma sample with extracted abundant proteins (lane P), showed that eluted protein from ProteoExtract Albumin/IgG removal kit resulted in a higher loading of proteins showing more bands (lane PD). This may be caused by the amount of eluted proteins being lower and also avoidance of masking by the presence of co-existing abundant proteins.

## Protein profile of samples concentrated from TCA precipitation and Vivaspin Turbo 4, 5 kDa ultrafiltration unit

Gels on Figs. 1B–1F were subsequently analyzed by Progenesis SameSpots software. The spots of those normalization volumes less than 35,317 were deleted at the filtering step. An extensive manual editing was performed after automated gel alignment and spot detection. The average number of detected protein spots was 600 spots for crude plasma samples precipitated with TCA, 604 spots for eluted protein samples precipitated with

TCA, 565 spots for eluted protein concentrated by Vivaspin Turbo 4, 5 kDa ultrafiltration unit, 495 spots for the bound protein precipitated with TCA and 659 spots for the bound protein with Vivaspin Turbo 4, 5 kDa ultrafiltration unit. There were 39 more spots on gels of TCA precipitated samples than the samples concentrated with Vivaspin Turbo 4, 5 kDa ultrafiltration unit, however, there were 164 more spots on the gels of bound protein samples concentrated by Vivaspin Turbo 4, 5 kDa ultrafiltration unit than the samples precipitated with TCA. Further, the resolution of the low-molecular weight spots was better on bound protein concentrated by Vivaspin Turbo 4, 5 kDa ultrafiltration unit. Thus, the method of protein samples concentrated by Vivaspin Turbo 4, 5 kDa ultrafiltration unit was selected in this study. Samples obtained from ProteoExtract Albumin/IgG removal kit were run onto 2-DE gels (Fig. 2). Samples from crude plasma without treatment generated an average of 457 spots in a 2-D gel. After removal of the abundant proteins, 553 of eluted protein spots were observed in a 2-D gel. There were about 582 of abundant protein spots observed in the gel. This work clearly showed the benefit of abundant protein removal to enhance the separation of low abundance proteins. There were about 60 spots matched in 2-D gels, for both eluted and abundant proteins.

## A good reproducibility of depleting abundant proteins using a ProteoExtract Albumin/IgG removal column

Results of the reproducibility of the of the Seppro spin column (Fig. 3) in depleting abundant proteins was instable. The average ratio of E/(E + B) × 100% was 37 ± 11 (%, $x$ ± SD, $n$ = 27) with a coefficient variation (CV) of 30%. The ratio was increasing with linear relationship of $R^2$ = 0.57 (see the relationship data in the Supplemental Material), which means the column's efficiency of depletion decreased. This could be caused by decreased binding affinity due to increased irreversible specific or nonspecific binding of antibody epitopes by protein from previously processed samples. This also reflected from the ratio of B/(E + B) × 100%, which decreased during the 27 depletion cycles. Four quality control samples were studied as a parallel control for every other six samples preparation. The detail information were supplied in Supplemental Material, in which eluted protein rate of QC1D–QC4D was increased gradually from 26.32%, 27.37%, 42.53% to 56.61%, respectively; and the rate of bound protein of QC1B–QC4B was decreased from 88.21%, 58.55%, 57.47% to 43.39% respectively. This study demonstrated the limitations of this column for repeated depletion and regeneration. The results did not agree with the manufacture's claims that the column can be used up to 100 times.

Compared the CV of 30% for the recovery rate of eluted proteins using Seppro spin column to the CV of using ProteoExtract Albumin/IgG removal kit which was only 7.1%, this also shows that the reproducibility among each depletion process in the second kit was very good. Furthermore, after the proteins were concentrated, the recovery rate, calculated by the amount (μg) in the eluted or bound fractions after concentration, compared to the amount of protein without concentration, was 81 ± 6% for eluted protein and 82 ± 8 for bound protein. There was also good inter-column reproducibility. The ProteoExtract™ removal kit provides a binding capacity of 0.7 mg IgG and/or 2 mg albumin per column, indicating a limitation of 30 μL plasma based on an albumin concentration of 7 g/dL.

Depletion of albumin and IgG from human serum samples was consistently higher than 70% without binding significant amounts of other serum proteins, and so a sample loaded onto a 2-D gel may be 3–4 times more concentrated. The manufacturer states that the "remarkable selectivity provided by the resins and the optimized design of the columns result in background binding of less than 10% to other plasma proteins". Another advantage of the product is the pre-filled disposable gravity-flow columns, which allow the parallel processing of multiple samples. The whole procedure takes about 30 min, in comparison to the long procedure of the Suppro column, which takes up to 12 h.

### Elution buffer containing 10% glycerol was efficient to elute bound protein from the ProteoExtract Albumin/IgG removal column

Another consideration of our method evaluation was to find an appropriate elution buffer to elute the proteins bound to the extraction column resin, because a biomarker of interest may be in the eluted or bound fraction. Thus efficient and complete removal of bound protein was of importance. The elution buffer was not provided in the kit. Because glycerol is often used as a cosolvent to inhibit protein aggregation during protein refolding (*Vagenende, Yap & Trout, 2009*), 25 or 10% glycerol was added into Laemmli buffer (62.4 mm Tris base, 2% SDS) in order to elute the bound protein. The efficiency of the two elution buffers was further assessed. A protein concentration of 3.5 µg/µl is important for loading onto 1-D and 2-D gels. When Laemmli buffer with 25% glycerol was used, the concentration of the eluted bound fraction was very slow and moreover, the total volume of the concentrated eluted protein fraction could not be reduced to 120 µL, a required volume in order to reach the ideal protein concentration of 3.5 µg/µl. This problem may have been caused by the high percentage of glycerol in the fraction. When Laemmli buffer containing 10% glycerol was chosen as elution buffer, a final concentration volume of 115 µl could be achieved.

## CONCLUSION

In summary, the present study explored the optimization of a method for the preparation of low abundance and abundant plasma proteins. The efficiency, selectivity and reproducibility of Seppro columns and ProteoExtract™ removal kits were evaluated. The Vivaspin Turbo 4, 5 kDa ultrafiltration unit gave the best concentration of eluted sample fractions when compared with TCA precipitation or a ReadyPrep 2-D cleanup Kit. Even though the results of using a Seppro column showed efficient separation of abundant and low abundance proteins, repeated re-use of the high cost antibody-based column was limited to 27 depletion-regeneration cycles before binding capacity of abundant proteins was gradually reduced. In our experience therefore, the column failed to achieve the specification of the manufacturer (100 re-use cycles with good reproducibility). Even though ProteoExtract™ removal kits removed only two abundant proteins, it could achieve a three times concentration of low abundance proteins loaded onto a gel. This improved the separation of lower abundant proteins, which was demonstrated in the separation of both 1-D and 2-D gels. Furthermore, the depletions using this column showed good reproducibility between individual columns, the CV being less than 10% in

both protein fractions. Using a 10% glycerol in Lamili buffer clearly improved the elution speed during the depletion process by the ProteoExtract Albumin/IgG column and also improved the efficiency of the evaporation of the concentrated samples. The optimized method of preparation of low/high abundant plasma proteins was: plasma was eluted through a ProteoExtract Albumin/IgG removal column, the elution contains the low abundant proteins; and the column was then further washed with elution buffer containing 10% glycerol, the elution contains the high abundant proteins. All elutions were further concentrated using Vivaspin® Turbo 4 5kDa ultrafiltration units for 1 or 2-D gel electrophoresis.

## ABBREVIATION

**E**      eluted (low abundance) proteins
**B**      bound (high abundance) proteins
**CV**     coefficient of variance

### Funding

Shaobo Zhou received research funding from the University of Bedfordshire.
John H. Beattie and Shaobo Zhou were funded by the Department of Health, research project (024/0043, previously N050017). John H. Beattie was also funded by the Scottish Government Rural and Environment Science and Analytical Services Division and a Horizon 2020 RISE grant MILEAGE (Project 734931). The funders had no role in study design, data collection and analysis, decision to publish, or preparation of the manuscript.

### Grant Disclosures

The following grant information was disclosed by the authors:
University of Bedfordshire.
Department of Health, Research Project: 024/0043 and N050017.
Scottish Government Rural and Environment Science and Analytical Services Division and a Horizon 2020 RISE grant MILEAGE Project: 734931.

### Competing Interests

The authors declare that they have no competing interests.

### Author Contributions

- Henian Yang performed the experiments, analyzed the data, prepared figures and/or tables, and approved the final draft.
- Guijie Wang performed the experiments, analyzed the data, prepared figures and/or tables, and approved the final draft.
- Tiantian Zhang performed the computation work, authored or reviewed drafts of the paper, and approved the final draft.

- John H. Beattie conceived and designed the experiments, analyzed the data, authored or reviewed drafts of the paper, and approved the final draft.
- Shaobo Zhou conceived and designed the experiments, analyzed the data, performed the computation work, authored or reviewed drafts of the paper, and approved the final draft.

## Ethics

The following information was supplied relating to ethical approvals (i.e., approving body and any reference numbers):

The Rowett Institute, University of Aberdeen provided ethics approval for the use of small animals (604012).

## Data Availability

The raw measurements are available in the Supplemental Files.

## Supplemental Information

Supplemental information for this article can be found online at http://dx.doi.org/10.7717/peerj-achem.6#supplemental-information.

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
