# Peer review of "Establishing an optimized method for the separation of low and high abundance blood plasma proteins"

_PeerJ Analytical Chemistry, doi:10.7717/peerj-achem.6_

## Round 0.1 · original submission · Major Revisions

Please address the reviewers' comments carefully.

Reviewer 1 ·

Basic reporting

Professional article structure, figures, tables. Raw data shared.

Experimental design

Research question well defined, relevant & meaningful. It is stated how research fills an identified knowledge gap.

Validity of the findings

All underlying data have been provided; they are robust, statistically sound, & controlled.

Additional comments

The manuscript titled “Establishing an optimized method for the preparation of low and high abundance blood plasma proteins” from Henian Yang et al tested the efficiency and reproducibility of a method for optimal separation of low and high abundant proteins in blood plasma. The authors are attempting to describe an appropriate conditions to separate low and high abundance plasma proteins from three different aspects. And then an optimized method is presented.
In my opinion, there are some points should be addressed as following.
(1) Although this paper provided the detailed protocol for three methods to concentrate the eluted plasma fraction and the bound proteins, no analysis of their differences and similarities among them were mentioned. The characteristics among them should be systematically clarified, which made it inconvenient for readers to understand these methods. And in the part of “Results and Discussion”, the reason of why samples treated with TCA precipitation did not generate good quality 2-D gels should be properly explained.
(2) The result of protein quantification in this study is only analyzed by the method of BCA protein assay. If there is another way to proof the final result, it will be more convinced for the quality of this manuscript.
(3) In the use of the Seppro spin column and the ProteoExtract Albumin/IgG removal kit to clarify the reproducibility of abundant protein depletion, the parallel control should be designed among the same spin columns.
(4) The figures in this article need to be more organized and accessible. For example, if the lanes in figure1A are divided clearly, it will be easier to understand.

Reviewer 2 ·

Basic reporting

The language is clear enough. Can be improved slightly in terms of grammar and flow. Intro and Background are clearly written. There are appropriate references in the literature. Figures are good quality. We failed to detect any issues with the images.

Experimental design

The research work is original to the best of my knowledge. Research question is meaningful and well defined but is not relevant owing to omission of LC-MS. The researchers identify a knowledge gap but failed to fill it completely. The investigation is of good technical and ethical standard. The methods have been sufficiently described.

Validity of the findings

All underlying data provided. No speculative statements encountered. Conclusions are well drawn and in line with experimental findings.

Additional comments

The authors are commended for a very useful article. They have provided detailed coverage of various commercial protein separation strategies. These even include repeated use performance of the same. We think this manuscript will definitely add to the field. However, some crucial concerns remain, as listed below.
After reading the manuscript, we suggest the title be appropriately modified. This reviewer feels preparation be replaced with separation in the title. The Abstract is quite confusing. This reviewer suggests adding 1-2 lines stating the best method/combination of methods for optimized preparation of low and high abundance plasma proteins.

---

## Round 0.2 · accepted · Accept

The authors have addressed the reviewers' concerns, and the manuscript appears to be ready for publication.

Reviewer 1 ·

Basic reporting

No comment.

Experimental design

No comment.

Validity of the findings

No comment.

Additional comments

In the revised manuscript, the authors addressed most of the major questions raised by Reviewers improving both the main structure and quality of the present paper. I have no further additional comments.

Reviewer 2 ·

Basic reporting

The language is clear enough. Intro and Background are clearly written. There are appropriate references in the literature. Figures are good quality. We failed to detect any issues with the images.

Experimental design

The research work is original to the best of my knowledge. Research question is meaningful and well defined but is not relevant owing to omission of LC-MS. The researchers identify a knowledge gap but failed to fill it completely. The investigation is of good technical and ethical standard. The methods have been sufficiently described.

Validity of the findings

All underlying data provided. No speculative statements encountered. Conclusions are well drawn and in line with experimental findings.

Additional comments

The authors are commended for a very useful article. They have provided detailed coverage of various commercial protein separation strategies. These even include repeated use performance of the same. We think this manuscript will definitely add to the field. I am satisfied with the author responses and recommending for the acceptance.